# Proteomics of Salt Gland–Secreted Sap Indicates a Pivotal Role for Vesicle Transport and Energy Metabolism in Plant Salt Secretion

**DOI:** 10.3390/ijms232213885

**Published:** 2022-11-11

**Authors:** Chaoxia Lu, Yuanyuan Zhang, Ping Mi, Xueying Guo, Yixuan Wen, Guoliang Han, Baoshan Wang

**Affiliations:** 1Shandong Provincial Key Laboratory of Plant Stress Research, College of Life Sciences, Shandong Normal University, Jinan 250014, China; 2Shandong Provincial Key Laboratory of Crop Genetic Improvement, Ecology and Physiology, Shandong Academy of Agricultural Sciences, Jinan 250100, China

**Keywords:** *Limonium bicolor*, proteomics, salt gland, secretion, vesicular transport

## Abstract

Soil salinization is one of the major factors restricting crop growth and agricultural production worldwide. Recretohalophytes have developed unique epidermal structures in their aboveground tissues, such as salt glands or salt bladders, to secrete excess salt out of the plant body as a protective mechanism from ion damage. Three hypotheses were proposed to explain how salt glands secrete salts: the osmotic hypothesis, a hypothesis similar to animal fluid transport, and vesicle-mediated exocytosis. However, there is no direct evidence to show whether the salt gland–secreted liquid contains landmark proteins or peptides which would elucidate the salt secretion mechanism. In this study, we collected the secreted liquid of salt glands from *Limonium bicolor*, followed by extraction and identification of its constituent proteins and peptides by SDS-PAGE and mass spectrometry. We detected 214 proteins and 440 polypeptides in the salt gland-secreted droplets of plants grown under control conditions. Unexpectedly, the proportion of energy metabolism-related proteins increased significantly though only 16 proteins and 35 polypeptides in the droplets of salt-treated plants were detected. In addition, vesicle transport proteins such as the Golgi marker enzyme glycosyltransferase were present in the secreted sap of salt glands from both control and salt-treated plants. These results suggest that trans-Golgi network-mediated vesicular transport and energy production contributes to salt secretion in salt glands.

## 1. Introduction

Soil salinization severely restricts global agricultural production and plant growth. The rapid expansion of saline–alkali land has become one of the most urgent problems in the world. In addition to naturally occurring salinized land, secondary soil salinization resulting from unreasonable irrigation, climate change, and seawater intrusion is also becoming more widespread, leading to a marked decrease in farmland and total grain production [1]. The growth of most land plants (named glycophytes or nonhalophytes) is significantly inhibited under salt stress, such as reduced plant biomass, delayed flowering, and altered root system architecture. A concentration of 50 mM NaCl will significantly inhibit the growth and development of glycophytes such as maize (*Zea mays*), soybean (*Glycine max*), and rice (*Oryza sativa*) [2,3]. More importantly, some plants (named halophytes) can grow and complete their entire life cycle on saline soils containing at least 200 mM NaCl. Only about 1% of all land plants are halophytes [4,5], and recretohalophytes, such as *L. bicolor* [5], and salt-dilution halophytes, such as *Suaeda salsa* [6,7] belong to the typical euhalophytes. Only recretohalophytes have developed unique structures that distinguish them from glycophytes and other halophytes: salt glands and salt bladders [5,8,9]. According to the review of Caperta et al. (2020), *Limonium* belongs to one of the families of Plumbaginaceae (non-core Caryophyllales) with typical salt glands, and the secretory cells are prismatic, columnar, or conical [10]. The salt gland of *L. bicolor,* which is a multicellular structure with 16 cells, has been observed by Feng et al. (2014) [1]. The main function of salt glands is to secrete excess salt from the plant body to the outside or to store the salt ions in salt bladders to protect plant cells from ion damage. Much attention has recently been paid to unraveling the mechanisms by which salt glands develop. Several genes homologous to those involved in trichome differentiation play a critical role in the development of salt glands [11,12]. The experimental results of Feng et al. (2014) confirmed that under 200 mmol/L NaCl treatment, the secretion rates of Na^+^ and Cl^−^ secreted by salt glands were significantly enhanced, but epidermal cells and stomata almost did not secrete ions [1]. However, few studies have attempted to dissect the biochemical basis of salt secretion from salt glands. It is of great significance to clarify the molecules participating in salt secretion by salt glands to better understand the salt tolerance mechanisms of recretohalophytes.

Between the 1950s and 1970s, three hypotheses were proposed to explain salt secretion in salt glands: the osmotic hypothesis, similar to animal fluid transport, and vesicle-mediated exocytosis [9,13,14,15]. The osmotic potential hypothesis was first proposed in *Limonium latifolium*, supported by the ultrastructure of *Distichlis spicata.* It suggests that ions actively accumulate in salt glands; when the salt gland pressure reaches its highest, ions are discharged from the secretory pores of the salt gland cells through the microdroplets formed periodically to relieve pressure [16,17]. Similar to the animal fluid transport hypothesis, a transfer system was proposed by Levering and Thomson (1971) [14]. Some studies support this hypothesis; salt secretion was significantly decreased after Na^+^-ATPase specific inhibitor (ouabain) treatment in *Chloris gayana* [18,19]. Regarding vesicle-mediated exocytosis hypothesis, proposed by Ziegler and Lüttge in 1967 [13], received support from *L. bicolor.* The study showed that electron-dense substances accumulated in the vesicles of salt glands [20,21]. The results of Flowers et al. (2019) also supported this hypothesis; they collected flux data of Na^+^ and Cl^−^ in halophytes based on the membrane area and weight of the transport organs [22]. Recent evidence has shown that vesicle transport may play an important role in the secretion from salt glands [9,20,23]. Indeed, previous studies showed that the number of vesicles in salt gland cells increases after salt treatment, together with the observation of vesicle fusion with the plasma membrane [1]. Upon treatment of *L. bicolor* leaves with the vesicle transport inhibitor brefeldin A (BFA), the salt secretion ability of leaf discs sharply decreased, the Golgi apparatus of the salt glands became deformed, and the trans-Golgi network (TGN) disappeared [23]. Although these observations all provide physiological and cellular support for the contribution of transport routes to salt secretion, no biochemical evidence exists.

During salt gland secretion, some water is also secreted together with salts [24,25]. The fluid secreted by salt glands contains various ions, such as Na^+^, K^+^, Ca^2+^, Mg^2+^, Cl^−^, NO_3_^−^, SO_4_^2−^, and PO_4_^3−^ [26], and the less common ions Cs^+^, Rb^+^, Br^−^, and I^−^ [27]. Shimony and Fahn (1968) speculated that pectic substances are present in the secretions of salt cedar (*Tamarix*) salt gland secretions [15]. Around the same time, Polak and Waisel (1970) also hypothesized that the secretion sap from *Aeluropus* contains many organic substances, including proteins and free amino acids [28,29]. However, the accumulation of protein in salt gland secretions has not been demonstrated, nor has the identity of these putative proteins been established. Here, we grew *L. bicolor* plants and collected the sap secreted by salt glands via the leaf disc method. We then extracted the proteins and peptides in the secretion liquid and analyzed them by SDS-PAGE and liquid chromatography-mass spectrometry. We determined the identity of all proteins and peptides and explored their possible functions using bioinformatics analysis to decipher the possible mechanism behind salt secretion.

## 2. Results

### 2.1. Characteristics of L. bicolor Salt Glands

Salt glands of *L. bicolor* possess spontaneous blue fluorescence from ferulic acid, a constituent of cutin [30]. The density of salt glands was stable, at about 17 glands in each 100× field of view of the microscope [31]. Most salt glands were located near the veins and were scattered over the leaf surface rather than in clusters. Salt glands had four fluorescent foci (Figure 1a,b) and four secretory pores (Figure 1c). Salt crystals were observed after the excretory pores of salt glands excreted liquid treated by salt treatment (Figure 1d).

### 2.2. SDS-PAGE Analysis of Proteins in the Secretion Liquids

The leaf disc method was used to analyze the proteins in the secretion liquids. In order to verify the effect of leaf disc injury on the protein secretion of the salt gland, we performed Evans blue staining on leaf discs treated with different treatments and found that isolated leaf discs only showed marginal cell damage, while different treatments did not affect the relative activity of salt gland cells (Appendix A). When the leaf discs were treated with 200 mM NaCl, the concentration of Na^+^, K^+^, and Cl^−^ in secretion sap secreted from salt glands were 4.08 μM and 4.64 μM, but the concentration of Na^+^ and Cl^−^ were 0.54 μM and 0.71 μM under the control, which were significantly lower than NaCl condition. While the concentration of K^+^ was 0.12 μM significantly decreased under NaCl condition compared with 0.55 μM under control condition (Appendix A). To determine whether proteins were secreted by salt glands, we separated the sap collected from salt glands using SDS-PAGE, followed by silver staining to detect proteins. Indeed, we detected many proteins in the secretion liquids from both control and NaCl-treated salt glands, with most proteins showing molecular weights ranging from 15 to 65 kDa (Appendix A). Unexpectedly, NaCl treatment led to a marked drop in the protein numbers and amounts.

### 2.3. Protein Identification by Mass Spectrometry

We determined the protein identity and their individual mass in the secreted liquids from salt glands using mass spectrometry. We observed that the protein mass in the secretion sap from control and NaCl-treated leaf discs was similar (Figure 2), with most proteins having a molecular weight of 30~50 kDa. However, we detected far more proteins in the secretion sap of control leaf discs than from NaCl-treated leaf discs, which was consistent with the SDS-PAGE results.

We identified 214 proteins and 440 polypeptides in the secretion sap of control leaf discs. Surprisingly, we detected only 16 proteins and 35 polypeptides in NaCl-treated sap (Figure 2a,c), suggesting that both the number of proteins and peptides were significantly lower after salt treatment (Figure 2b).

### 2.4. Annotation of Proteins and Peptides

To explore the possible functions of the proteins and peptides detected in the secreted liquid of salt glands, we subjected them to Gene Ontology annotation. We thus determined that the proteins in the secreted sap of control leaf discs have different subcellular localizations, such as chloroplasts, mitochondria, peroxisomes, the cytosol, and the Golgi apparatus. About 17% of all control proteins localized to chloroplasts, 26% in cytoplasm, 7% in mitochondria, and 5% in peroxisomes (Figure 3). In addition, 38% of the proteins were uncharacterized or unknown. We noticed several organelle marker enzymes such as glycosyltransferases (Golgi marker), catalase (peroxisome marker), and cytochrome oxidase (mitochondrion marker) among salt gland-secreted proteins.

To deeply explore the potential functions and differences of proteins and peptides secreted by salt glands under salt stress and control conditions, we took a closer look at the possible functions of proteins in secreted sap. For control leaf discs, 40 of the identified proteins were involved in photosynthesis. Of these, 36 proteins were not present in the sap of NaCl-treated salt glands, with three Rubisco large chains and one Rubisco small chain subunit remaining (Appendix A). In addition, we detected 26 proteins involved in respiration in the sap of control leaf discs, of which only four (two ATP synthase alpha subunits, one ATP synthase beta subunit, and one xanthine dehydrogenase) remained in the sap of NaCl-treated salt glands (Appendix A).

After bioinformatics analysis, proteins related to vesicle transport were mainly involved in cytoskeleton-associated proteins, protein kinases, and proteins with functional domains (Table 1). Proteins related to photosynthesis and energy metabolism accounted for 30.8% of the number of total proteins secreted in the control group, while vesicle transport-related proteins accounted for 3.7% of the total number of proteins secreted in the same samples. Proteins related to photosynthesis and energy metabolism accounted for 50% of the total number of proteins secreted by leaf discs exposed to salt stress. Notably, enzymes related to reactive oxygen species (ROS) scavenging, such as peroxidase, superoxide dismutase, and catalase, were no longer detected in the sap from salt-treated salt glands.

We also identified various peptides in the secreted sap of salt glands, which are both degradation products and peptides with biological functions. Thirty-eight peptides with potential functions were randomly chosen for annotation (Table 2). These peptides were mainly derived from photosynthesis-related enzyme chains, elongation factor fragments, ATP synthase subunits, and special domain fragments, which play an important role in plant growth and development, apoptosis, signal transduction, metabolism, and response to stress (Table 2). Elongation factor 1-alpha, phosphoglycerate kinase (fragment), ATP synthase (alpha and beta subunits), and phosphoglycerate kinase were related to the salt stress response. Similar to the protein results, the number and type of peptides decreased significantly under 200 mM NaCl treatment. However, the Ribulose bisphosphate carboxylase large chain, as well as the alpha and beta subunits of ATP synthase, were still present in sap after salt treatment.

## 3. Discussion

*L. bicolor* has special salt-secreting structures on its leaf surface called salt glands. In this study, we showed that the liquid secreted from these salt glands contained a variety of proteins and polypeptides, whose number and variety sharply declined under salt stress compared to control conditions. We believe that the presence of peptides of secretion sap under control conditions may be due to a sudden enhancement of vesicular transport carrying a large number of immature protein fragments. However, under 2 weeks of salt condition, the *L. bicolor* could adapt the activity of salt glands to secrete Na^+^ and Cl^−^. We conclude that the decrease in the number of proteins and peptides is related to the adaptation of the salt gland secretion salt of the halophytes *L. bicolor* under salt treatment, and the remaining proteins and peptides after 2 weeks of salt treatment may be involved in the normal salt secretion under salt stress. According to our result, these proteins and polypeptides participate in vesicular transport, energy production, and material metabolism. In order to better understand the role of proteins and peptides in salt-gland secretion, we tried to discuss the functions of vesicular transport-related proteins, energy metabolizing proteins, cargo proteins, stress-related proteins, and their roles in salt-gland secretion. In particular, we identified Golgi marker enzymes, such as glycosyltransferases and vesicle transport cargo proteins, among the secreted proteins. These results indicate that vesicle transport plays an important role in salt gland secretion in the recretohalophyte *L. bicolor* and that many salt secretion-unrelated metabolic processes, such as ATP production, are put on hold or reduced under high-salinity conditions.

As a conventional membrane transport pathway, the steps of vesicular transport (budding, transport, tethering, docking, and fusion) engage several organelles and require the participation of multiple protein complexes [49,50]. The TGN coordinates cell organs by receiving, secreting, and directing vesicles to their intended destinations [51,52]. Salt secretion from salt glands was recently linked to vesicle transport. For instance, Smaoui et al. (2011) undertook a lanthanum tracing study in the salt glands of the recretohalophyte saltbush (*Atriplex*), which revealed that lanthanum is deposited in endocytic vesicles at the nuclear membrane, in the rough endoplasmic reticulum, and the Golgi lumen [53]. After treatment of *L. bicolor* leaf discs with the vesicle transport inhibitor BFA, the Golgi apparatus was reported to be deformed, the *cis*-Golgi apparatus disappeared, so-called BFA compartments formed in the cytoplasm, and the secretion capacity of leaf discs dropped sharply. A threshold concentration of BFA resulted in almost no fluid secretion by leaf discs [23]. BFA is a fungal-derived macrolide that specifically inhibits Golgi-mediated secretion and is a powerful tool for dissecting the plant secretion and endophagocytic pathway [54,55,56]. These studies indicate that the Golgi apparatus and TGN play a key role in salt gland secretion. Hamaji et al. (2009) found that vesicles were more active in Arabidopsis (*Arabidopsis thaliana*) under salt treatment, with salt ions not only accumulating in vacuoles but also in these active vesicles [57]. This report also provided evidence for the participation of vesicle transport in salt ion movements [57]. Our experimental results showed for the first time that the secretions of salt glands contain proteins and peptides, especially the Golgi marker protein glycosyltransferase, as well as enzymes associated with energy metabolism (Table 1 and Appendix A).

Glycosyltransferases are usually located at the Golgi membrane or lumen [58]. The presence of glycosyltransferases and other proteins in the secreted liquid may reflect a lack of selectivity in vesicle transport. The selectivity of ion channels and transporters depends on the structural properties of ion channel and transporter proteins; however, the vesicle transport depends on the local solute concentration near the site of vesicle formation and the charge of the vesicle membrane and is not selective [22,59]. In addition, we also detected many cargo proteins in the secreted liquid transported by vesicles, such as chitinases and xyloglucan endotransglucosylases/hydrolases (XTHs). This observation also suggests that salt glands may be involved in a rich vesicle transport.

We identified many energy-related proteins in the secretions, which supports the notion that vesicle transport activities of salt secretion need a lot of energy. Vesicle transport mainly assists in the transport and exchange of substances (such as proteins and lipids) between the organellar membranes [60]. Here, we found that the liquid from salt gland secretions also contained proteins related to photosynthesis and energy metabolism (30.8% of all proteins), ROS metabolism, protein synthesis, and some uncharacterized proteins (21.5%). Most of these proteins were detected in the form of polypeptides and originated from various organelles. The secreted proteins included Ribulose bisphosphate carboxylase and ATPase (Appendix A), indicating that large amounts of energy and an H^+^ gradient are required to sustain salt secretion from salt glands, with abundant photosynthesis producing multiple metabolic substrates. Many reports have also shown that the many mitochondria and Golgi in the salt glands provide the necessary energy for salt secretion [1,61].

Our results showed that the control leaf discs of *L. bicolor* plants secreted many proteins after floating on 200 mM NaCl solution for 24 h, which was profoundly different from those of the plants watered with 200 mM NaCl solution for 2 weeks. After 2 weeks of treatment with 200 mM NaCl, the local ion concentration is known to increase near the vesicles, as does the speed of vesicle movement [1,57]. After 2 weeks of salt treatment, plants adapted to salt stress; the vesicles perform their normal transport function within the cell while also secreting ions (Appendix A). Compared to salt secretion by salt glands under normal conditions, the secreted liquid after salt treatment contained 16 proteins, which were divided into three main categories: proteins related to photosynthesis and respiration (50%), protein synthesis, and uncharacterized proteins (18.8%). Moreover, the proportion of photosynthesis- and energy metabolism-related proteins increased significantly in salt-stressed samples compared to control samples, indicating that the increased salt secretion exhibited by salt-stressed leaf discs may require more energy expenditure than control conditions. This higher energy quota may be the result of normal vesicle transport after adaptation to salt stress.

In addition, we identified several proteins typically associated with stress responses in the control samples, such as ROS-scavenging enzymes, chitinases, and XTHs. Notably, we did not detect these proteins upon salt treatment. We believe that ROS-scavenging enzymes may be related to plant response to salt stress and resistance to oxidative stress in the leaf cytoplasm, so it is kept inside the plant cells instead of being secreted [62]. XTHs also respond to biotic/abiotic stresses. Forty-two *SeXTH* genes were upregulated in the succulent halophyte glasswort (*Salicornia europaea* L.) under salt and drought stress [63,64]. We also identified elongation factor 1-alpha (EF1A), an indicator of cell growth and development, which regulates protein synthesis by altering the expression of eEF1A to rapidly respond to external environmental stress and cellular signals [65]. We detected several uncharacterized proteins in samples from both control and salt-treated leaf discs; more attention should therefore be paid to the functional dissection of these uncharacterized proteins secreted by plant salt glands, as they may contribute to the regulation of salt secretion or salt gland formation. Many known polypeptides contribute to plant responses to salt stress. The Arabidopsis homolog to tomato CAP-derived peptide 1 (CAPE1) and Cysteine-rich transmembrane module3 (CYSTM3) raises salt sensitivity and negatively regulates salt tolerance by inhibiting Na^+^ efflux in *Arabidopsis* roots [66,67].

In addition, we detected fragments from the large subunit of Ribulose bisphosphate carboxylase, as well as the alpha and beta subunits of ATP synthase, under both control and salt treatment, hinting at their role in maintaining active salt secretion and providing energy under prolonged salt stress. These results also suggest that plants respond quickly to salt stress through functional peptides when suddenly exposed to salt stress. Since vesicle transport is determined by the local ion concentration near the vesicle and the charge of the vesicle membrane, the complement of secreted proteins appeared to be poorly reproducible (Figure 2). We detected the Golgi marker enzyme glycosyltransferase in the secretion sap, which confirmed that salt ions were excreted in vivo through the classical secretion pathway, with the involvement of TGN-mediated vesicle transport in salt glands. Lu et al. (2020) showed that the SYNTAXIN OF PLANTS61 (SYP61) protein located at the TGN plays an important role in salt secretions from salt glands via virus-induced gene silencing of its encoding gene [23]. Our results provide support for the endocytosis hypothesis in the secretion mechanism by salt glands [5,23,28]. However, which proteins participate in the secretion of salt ions, as well as their mode of action, needs to be studied in more detail. On the other hand, under salt treatment after two weeks, the Na^+^ and Cl^−^ concentration in secretion sap significantly increased, compared with the amount of protein and peptides (Figure 2 and Appendix A). We speculated that the main function of the salt gland is secreting ions, and this kind of ion and protein secretion balance is how to control under salt conditions. Whether there are proteins that regulate salt gland secretion, we do not yet know.

## 4. Materials and Methods

### 4.1. Plant Materials and Culture Conditions

*L. bicolor* seeds were collected from the Yellow River Delta, Dongying, China (37°20′ N; 118°36′ E) in 2020 and stored in a refrigerator at 4 °C for 6 months before use. Healthy, uniform seeds were selected and planted in a Petri plate with 30 mL of Murashige and Skoog (MS) medium. After germinating for 7 days (22 ± 3 °C), seedlings were transplanted into well-mixed soil (soil: vermiculite: perlite, 3:1:1, *w*/*w*/*w*) and grown in a greenhouse (28 ± 3 °C/23 ± 3 °C, day/night) at a light intensity of 600 mol/m^2^/s (with a 16-h-light/8-h-dark photoperiod) and 70% relative humidity. When the sixth true leaf of the seedling grew, uniform seedlings were used for further experiments. The control plants were irrigated with Hoagland’s nutrient solution containing 5 mM KNO_3_, 5 mM Ca(NO_3_)_2_, 2 mM MgSO_4_, 1 mM KH_2_PO_4_, 20 μM Fe-EDTA, 46 μM H_3_BO_3_, 9.1 μM MnCl_2_, 0.32 μM CuSO_4_, 0.76 μM ZnSO_4_, and 0.11 μM Na_2_MoO_4_, and the pH was adjusted to 6.2 ± 0.1 using 1 M KOH and H_2_SO_4_. The treatment plants were watered with Hoagland’s nutrient solution containing 200 mmol/L NaCl with an increment of 50 mmol/L every 24 h. After reaching the final concentration, the plants were watered every three days and treated for two weeks.

### 4.2. Salt Gland Observations

Salt glands were observed using the method described by Li et al. (2020) [68]. Leaves were observed under a scanning electron microscope according to the method of Shah et al. (2018) [69]. The true leaves of *L. bicolor* seedlings were placed in enzymolysis solution and shaken at 240 g for 2 h in the dark at 27 °C. Then, 1% (*w*/*v*) diselase, 0.1% (*w*/*v*) pectolyase Y-23, and 1% (*w*/*v*) cellulase Y-C were dissolved in PH 5.7 buffer (20 mM MES, 20 mM KCl, 0.4 M mannitol) to make enzymolysis solution. The leaves were transferred to a homogenizer containing phosphate-buffered saline of pH 7.4 for grinding. The homogenate was filtered through a 70-µm cell screen and stored on ice. The salt glands were observed under UV excitation (330–380 nm) using differential interference contrast microscopy (DIC). In addition, we calculated the ions concentration of secretion sap under 200 mM NaCl treated and control conditions using the method of Lu et al. (2020) [23]. Moreover, the relative activity of cells was measured using the method of Gaff et al. (1971) [70].

### 4.3. Extraction of Proteins from the Secretion Liquid of Salt Glands

Uniform *L. bicolor* leaves (usually the sixth true leaf) were used to collect the secretion liquid, as described by Lu et al. (2020) [23] (Appendix A). Leaf discs without a main vasculature were collected with a hole puncher (diameter 1 cm), and each leaf disc was placed in 200 mM NaCl solution for 24 h in a growth chamber, with the abaxial surface facing up. Mineral oil was layered onto the leaf discs immediately after floating on the NaCl solution. After 24 h, many droplets formed on the leaf disc surface and were collected with a micropipette. The collected secretion liquid was centrifuged at 4000× *g* for 10 min at room temperature. The lower aqueous phase was used to extract the proteins by trichloroacetic acid (TCA)-acetone precipitation after removing the upper mineral oil phase. The aqueous phase was divided into 10-mL centrifuge tubes, and three volumes of pre-cooled TCA-acetone solution (20%, *w*/*v*) were added. After fully mixing, the samples were incubated at −20 °C for 2 h to precipitate proteins, which were collected using centrifugation at 10,000× *g* at 4 °C for 10 min. The pellets were then rinsed twice with pre-cooled acetone and air-dried. The pellet was then resuspended in 2× loading buffer before SDS-PAGE. Protein separation was carried out according to Laemmli. (1970) [71]. A PAGE gel quick preparation kit (10%) was used to prepare the gels (Yeasen, Shanghai, China). The separation gel consisted of 4 mL of 10% separation gel buffer, 4 mL of 10% separation gel solution, and 70 μL of 10% modified ammonium persulfate solution. The stacking gel comprised 1 mL of 10% color-concentrated gel buffer, 1 mL of 10% concentrated gel solution, and 18 μL of 10% modified ammonium persulfate solution. A protein silver stain plus kit was used to stain proteins (SK6020, Coolaber, Beijing, China).

### 4.4. Protein Identification, Annotation, and Characterization

The protein analysis was performed with liquid chromatography-mass spectrometry (Q Exactive HF-X, Thermo Fisher, Waltham, MA, USA). The samples were separated using a nano liquid chromatographic system with a flow rate of 5–300 nL/min (UltiMate 3000 RSL, Thermo Fisher, USA). The protein samples were dissolved in loading buffer, injected with an automatic sampler, bound to a C18 capture column (3 μm, 120 A, 100 μm × 20 mm), and then eluted into an analysis column (2 μm, 120 A, 750 μm × 250 mm) for separation. An analytical gradient was established using two mobile phases (mobile phase A: 3% [*v*/*v*] DMSO, 0.1% [*v*/*v*] formic acid, and 97% H_2_O; and mobile phase B: 3% [*v*/*v*] DMSO, 0.1% [*v*/*v*] formic acid, and 97% acetonitrile [ACN]). The flow rate of the liquid phase was set to 300 nL/min. The MS analysis was performed in DDA mode, with each scan cycle consisting of a full MS scan (R = 60 K, AGC = 3 × 10^6^, Max IT = 20 ms, scan range = 350–1800 *m*/*z*), followed by 20 MS/MS scans (R = 15 K, AGC = 2 × 10^5^, max IT = 100 ms).

### 4.5. Data Analysis

The mass spectrometry data were retrieved by MaxQuant (V1.6.2.10) using the MaxLFQ retrieval algorithm [72]. The retrieval database was the Proteome Reference database of Plumbaginaceae in UniProt. Protein fragments or subunits with molecular weight less than 10,000 Da were selected for collation and functional annotation.

## Figures and Tables

**Figure 1 ijms-23-13885-f001:**
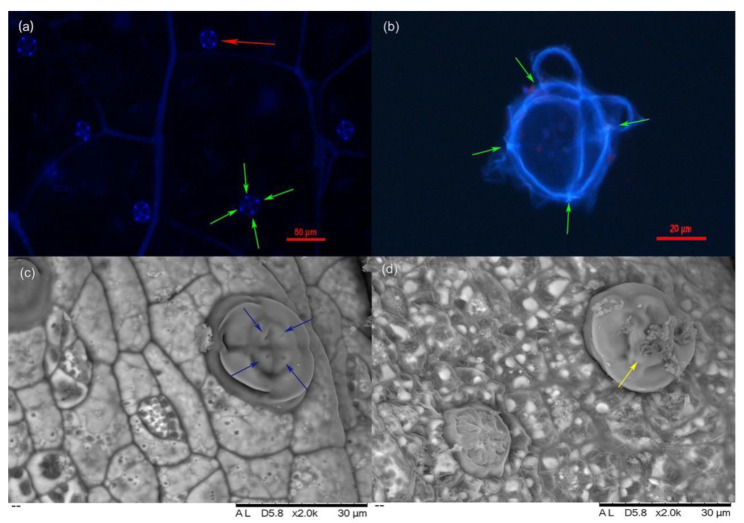
Salt gland of *L. bicolor* under DIC and scan electron microscope. (**a**) Salt gland observed under DIC microscope at 330–380 nm UV light; (**b**) Salt gland after enzymatic hydrolysis under DIC microscope at 330–380 nm UV light; (**c**) Salt gland observation by scan electron microscope under normal conditions; (**d**) Salt gland observation by scan electron microscope under 200 mM NaCl treatment for 7 days; Red arrow indicates salt gland and green arrows in (**a**,**b**) show four luminous spots of salt gland. Blue arrows indicate secretory pores, and yellow arrow indicates salt crystals.

**Figure 2 ijms-23-13885-f002:**
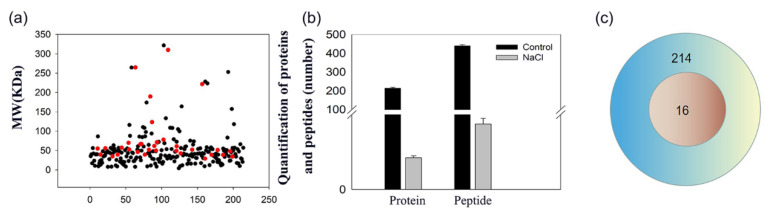
Molecular mass distribution and number of proteins in salt gland secretion sap of *L. bicolor.* (**a**) Black points are the control group, red points are the salt treatment group; ordinate is the molecular mass of the protein, and abscissa is the protein number; (**b**) Black column represents the control group, and the gray represents 200 mM NaCl treatment; (**c**) Venn pattern of proteins in salt gland secretion sap of *L. bicolor.* The blue circle shows the protein number of control, and the red circle shows the number of proteins under 200 mM NaCl treatment.

**Figure 3 ijms-23-13885-f003:**
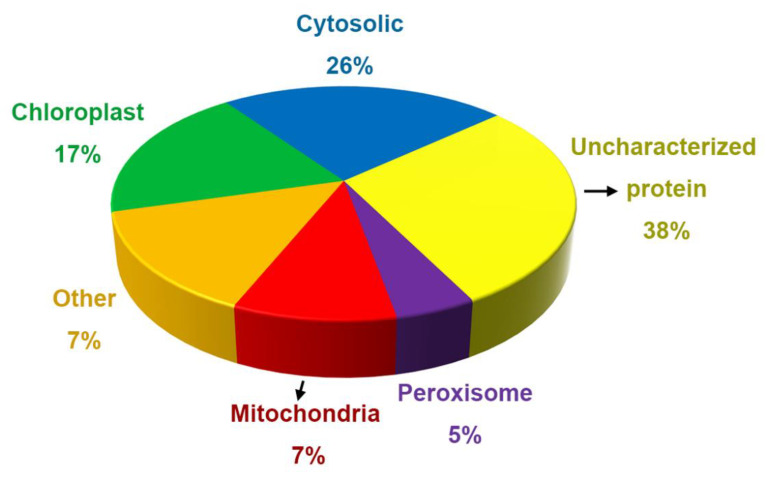
Percentage of the proteins in subcellular organelles in the salt gland secretion fluid of *L. bicolor* plants without NaCl treatment.

**Table 1 ijms-23-13885-t001:** Vesicle transport-related proteins in the secreted sap of *L. bicolor* salt glands treated with 0 and 200 mM NaCl.

Accession	Description	MW (kDa)	Control	NaCl
A0A0J8DU34	Tubulin_C domain-containing protein (Fragment) OS = *Beta vulgaris* subsp. *vulgaris*	31.849	+	−
A0A1L5JKA9	Actin 11 OS = *Sesuvium portulacastrum*	41.67	+	−
A0A1L5JKC2	Tubulin alpha chain OS = *Sesuvium portulacastrum*	49.656	+	−
M4WH59	Actin 1 OS = *Dionaea muscipula*	41.721	+	−
A0A0K9Q9G9	Protein kinase domain-containing protein (Fragment) OS = *Spinacia oleracea*	24.543	+	−
A0A0K9QDZ4	Protein kinase domain-containing protein OS = *Spinacia oleracea*	108.3	+	−
A0A0K9QE47	GTD-binding domain-containing protein OS = *Spinacia oleracea*	97.611	+	−
Q9MB82	Plasma membrane H+-ATPase (Fragment) OS = *Nepenthes alata*	20.592	+	−

**Table 2 ijms-23-13885-t002:** Peptides analysis and functional annotation in the secreted sap of *L. bicolor* salt glands treated with 0 and 200 mM NaCl.

Accession	Description	MW (kDa)	Control	NaCl	Function
A0A0F7G6K5	Ribulose bisphosphate carboxylase large chain (Fragment) OS = *Paronychia herniarioides*	47.9	+	+	Influence the Vmax of carboxylation and CO_2_/O_2_, specificity factor in *Chlamydomonas reinhardtii* [32]
A0A068ELN9	Ribulose bisphosphate carboxylase large chain (Fragment) OS = *Limonium dragonericum*	50.151	+	+	
A0A3S9XK16	Ribulose bisphosphate carboxylase large chain (Fragment) OS = *Plumbago zeylanica*	46.635	+	−	
A0A411JQT5	Ribulose bisphosphate carboxylase large chain OS = *Dioncophyllum thollonii*	54.893	+	−	
A0A650AK20	Ribulose bisphosphate carboxylase large chain OS = *Limonium aureum*	52.67	+	+	
F2WAU2	Ribulose bisphosphate carboxylase large chain (Fragment) OS = *Tecticornia disarticulata*	49.537	+	−	
I3XLX8	Ribulose bisphosphate carboxylase large chain (Fragment)	45.152	+	−	
I6WY50	Ribulose bisphosphate carboxylase large chain (Fragment) OS = *Limonium sinense*	46.023	+	−	
O47047	Ribulose bisphosphate carboxylase large chain (Fragment) OS = *Eriogonum flavum*	49.73	+	−	
P28410	Ribulose bisphosphate carboxylase large chain (Fragment) OS = *Drosera petiolaris*	48.899	+	−	
A0A0H3Y5V0	Ribulose-1,5-bisphosphate carboxylase/oxygenase large subunit (Fragment) OS = *Delosperma peglerae*	8.7523	+	−	Covalently linked to form dimer under oxidative stress and reduce photosynthesis in tobacco [33]; triggered in response to the exposure of low irradiance condition and reduce the activity of Rubisco in Arabidopsis [34]
A0A140ETK2	Ribulose-1,5-bisphosphate carboxylase/oxygenase large subunit (Fragment) OS = *Aerva lanata*	17.336	+	−	
A0A346TN31	Ribulose-1,5-bisphosphate carboxylase/oxygenase large subunit (Fragment) OS = *Atriplex vesicaria*	9.2556	+	−	
Q6A161	Ribulose 1,5 bisphosphate carboxylase/oxygenase, large subunit OS = *Limonium gibertii*	53.413	+	−	
A0A410S7A2	Ribulose-1,5-bisphosphate carboxylase/oxygenase large subunit (Fragment) OS = *Achyranthes aspera*	12.178	+	−	
G0WXT9	Ribulose-1,5-bisphosphate carboxylase/oxygenase large subunit (Fragment) OS = *Amaranthus deflexus*	33.481	+	−	
A0A0J7YLM1	Elongation factor Tu (Fragment) OS = *Beta vulgaris* subsp. *vulgaris*	44.306	+	−	Proved to be an independent structural and functional unit of Tryptic fragment and could bind with GDP in *Escherichia coli* and rabbit [35,36]
A0A0K9Q7G6	Elongation factor Tu OS = *Spinacia oleracea*	49.602	+	−	
A0A0K9Q895	PCI domain-containing protein (Fragment) OS = *Spinacia oleracea*	43.288	+	−	Could be believed have catalytic activity [37]
A0A0K9Q9G9	Protein kinase domain-containing protein (Fragment) OS = *Spinacia oleracea*	24.543	+	−	Protein biosynthesis; signal transmission [38]; proved to response to biotic and abiotic stress [39]
A0A0K9QJ83	Ku domain-containing protein (Fragment) OS = *Spinacia oleracea*	93.807	+	−	DNA-binding domain [40]
A0A0K9RD36	NB-ARC domain-containing protein (Fragment) OS = *Spinacia oleracea*	105.01	+	−	Regulate plant disease resistance protein [41]
A0A249Y703	Elongation factor 1-alpha OS = *Hylocereus polyrhizus*	49.869	+	−	As molecular chaperone under salt stress in yeast and plants [42]; play a role of response to heat stress in wheat [41]
A0A5B8WHU7	Elongation factor 1-alpha OS = *Hylocereus undatus*	49.373	+	−	
A0A411JPW9	ATP synthase subunit beta OS = *Ceratostigma willmottianum*	53.874	+	+	As a plant cell death regulator [43]; response to waterlogging stress in barley (*Hordeum vulgare* L.) [44]; response to heat stress in rice (*Oryza sativa* L.) [45]; response to salt and drought stress [46]
A0A411JS23	ATP synthase subunit beta OS = *Limonium tenellum*	53.218	+	−	
G8A3M7	ATP synthase subunit beta (Fragment) OS = *Alluaudia procera*	50.753	+	−	
A0A0K9RQ86	ATP synthase subunit beta OS = *Spinacia oleracea*	59.387	+	−	
B0L803	33 kDa OEC protein (Fragment) OS = *Salicornia veneta*	18.289	+	−	Related to the recovery of high-Light stress [47]; response to heat stress in *Zea mays* L.
G4WNY3	NAD(P)H-quinone oxidoreductase subunit 5, chloroplastic (Fragment) OS = *Stegnosperma halimifolium*	57.819	+	−	Alleviated photosynthetic inhibition in tobacco leaves under Cd stress [48]
G5DVX3	Phosphoglycerate kinase (Fragment) OS = *Silene latifolia*	51.21	+	−	Could be believed response to salt stress [46]
A0A411JPX2	ATP synthase subunit alpha OS = *Ceratostigma willmottianum*	55.662	+	+	Response to salt and drought stress [46]
A0A411JPX2	ATP synthase subunit alpha OS = *Ceratostigma willmottianum*	55.662	+	+	Response to salt and drought stress [46]
A0A411JRY5	ATP synthase subunit alpha OS = *Limonium tenellum*	55.584	+	−	
A0A411L2V2	ATP synthase subunit alpha, chloroplastic OS = *Froelichia latifolia*	55.562	+	−	
D3WEW2	ATP synthase subunit alpha OS = *Plumbago auriculata*	55.647	+	+	
H8Y672	ATP synthase subunit alpha OS = *Silene vulgaris*	58.181	+	−	
P29409	Phosphoglycerate kinase, chloroplastic (Fragment) OS = *Spinacia oleracea*	45.572	+	−	Response to salt stress [46]

## Data Availability

The data and materials that were analyzed in the current study are available from the corresponding author upon reasonable request.

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
