# Peer review of "Proteomics of Salt Gland–Secreted Sap Indicates a Pivotal Role for Vesicle Transport and Energy Metabolism in Plant Salt Secretion"

_ijms, 2022, doi:10.3390/ijms232213885_

Round 1
Reviewer 1 Report
General
The results are novel but the question of whether the results could be artefacts of the experimental system is not discussed, nor the reasons for the decrease in numbers of peptides found after salt treatment. Evidence of salt secretion is not provided and some relevant literature is missed.
Specific comments
Lines 30-40. The opening paragraph is unnecessary in the context of this paper and could be deleted. There are no new facts here and the paper is not about plant breeding but the physiology of salt glands. Better to concentrate on the latter.
Lines 41-42. “have lost their salt tolerance mechanisms during long-term evolution and are salt-sensitive (named nonhalophytes”. It has been argued that higher plants evolved from fresh-water species and so salt-tolerance is a secondary trait. The text should be adjusted as it has not been established that plants have “lost their salt tolerance”. Why is the term “nonhalophytes” used rather than the more commonly used ‘glycophytes’?
Lines 47-48. Allocating halophytes to three categories is misleading. Recretohalophytes generally fall within the category of euhalophytes. ‘pseudohalophytes’ are only mentioned in reference number 12 (and not in numbers 10 or 11) where they are defined as “pseudohalophytes (i.e., salt-excluding halophytes).”. This term is misleading as all halophytes restrict the quantity of saline ions that they take up and, in the sense widely misused in the literature, are excluders.
Line 59. Since the paper is about the mechanism of secretion of ions, the three hypotheses should be described in more detail. Summarise the three hypotheses and comment on how they might be distinguished experimentally.
Lines 61-62. The authors do not mention recent reviews of Caperta et al (2020) where the origin of and function of glands within the Plumbaginaceae is discussed or, particularly relevant, the review of Flowers et al (2019), in which vesicular transport in halophytese is discussed.
Caperta AD, Rois AS, Teixeira G, Garcia-Caparros P, Flowers TJ (2020) Secretory structures in plants: lessons from the Plumbaginaceae on their origin, evolution and roles in stress tolerance. Plant Cell and Environment 43 (12):2912-2931. doi:10.1111/pce.13825
Flowers TJ, Glenn EP, Volkov V (2019) Could vesicular transport of Na+ and Cl- be a feature of salt tolerance in halophytes? Annals of Botany 123 (1):1-18. doi:10.1093/aob/mcy164
Line 84 et seq. As this paper is about salt-secreting glands, the authors should include, at a minimum, concentrations of Na, K and Cl in the external solution on which the leaf disks were floated and in the secretions collected from the glands in order to show the glands were functioning in the way that might be expected/has been previously demonstrated.
Line 111 et seq. Provide an image of the droplets on the leaf surface.
Line 114. What was the size of discs?
Line 173 Figure 2. This figure is not of very good quality and does no more than show proteins were present and could be separated. I suggest the figure is transferred to the supplementary data. The text (lines 166-175) should be adjusted as appropriate.
Line 202-203. “malate dehydrogenase (mitochondrion marker).” There are malate dehydrogenases other than in the mitochondria: e.g., Oh et al (2016).
Oh SJ, Gu DR, Jin SH, Park KH, Lee SH (2016) Cytosolic malate dehydrogenase regulates RANKL-mediated osteoclastogenesis via AMPK/c-Fos/NFATc1 signaling. Biochem Biophys Res Commun 475 (1):125-132. doi:10.1016/j.bbrc.2016.05.055
Line 242, 243. “liquid secreted from these salt glands contained a variety of proteins and polypeptides”: Why are there so many different polypeptides? No explanation is provided; nor is possible contamination from cut cells discussed. Does the size of the disc (and so ratio of cut edge to surface area) affect the number of proteins found in the collected drops? Can droplets be collected from intact leaves to eliminate possible leakage from damaged cells?
Line es 243, 244. “whose number and variety sharply declined under salt stress compared to control conditions.”. Why should salt treatment decrease the number of proteins? This needs comment in the Discussion.
Minor comments
Line 21 Clarify “number proportion”.
Line 38/9. “valorization”. Delete
Line 59 “have been” ‘were’.
Line 124 “rinsedtwice” ‘rinsed twice’
Line 157 “Salt crystals were excreted”. ‘Crystals’ were surely not excreted. but form after excretion of liquid containing salts.
Line 206: Figure 4. Spelling of “peroxisome” perooxisome’.
Line 208 “saps”; ‘sap’.
Author Response
Dear Editor and Reviewers,
Thank you very much for your suggestions and critical comments on our paper submitted to International Journal of Molecular Sciences. We have revised the manuscript according to the comments and suggestions. Attached, please find the revised version. The main revised parts are marked in blue in the paper. The following is a detailed explanation how we have complied with the reviewers’ suggestions.
Reviewer1
Comments and Suggestions for Authors
General
The results are novel but the question of whether the results could be artefacts of the experimental system is not discussed, nor the reasons for the decrease in numbers of peptides found after salt treatment. Evidence of salt secretion is not provided and some relevant literature is missed.
Specific comments
(1)Lines 30-40. The opening paragraph is unnecessary in the context of this paper and could be deleted. There are no new facts here and the paper is not about plant breeding but the physiology of salt glands. Better to concentrate on the latter.
Response: Thank you for your critical comments. We have deleted the sentences to reduce the introduction of background.
(2)Lines 41-42. “have lost their salt tolerance mechanisms during long-term evolution and are salt-sensitive (named nonhalophytes”. It has been argued that higher plants evolved from fresh-water species and so salt-tolerance is a secondary trait. The text should be adjusted as it has not been established that plants have “lost their salt tolerance”. Why is the term “nonhalophytes” used rather than the more commonly used ‘glycophytes’?
Response: Thank you for your critical comments. We have modified the description of nonhalophytes. And we choose the more appropriate “glycophytes” to replace “nonhalophytes”. We agree that salt tolerance play a secondary role in plant evolution. We have redescribed the growth inhibition of glycophytes under salt stress. And more importantly, we highlight the advantages of halophytes growing under salt stress, which has greater research potential. And the details were shown on lines 39-41, 43 and 49 of the revised version.
(3) Lines 47-48. Allocating halophytes to three categories is misleading. Recretohalophytes generally fall within the category of euhalophytes. ‘pseudohalophytes’ are only mentioned in reference number 12 (and not in numbers 10 or 11) where they are defined as “pseudohalophytes (i.e., salt-excluding halophytes).”. This term is misleading as all halophytes restrict the quantity of saline ions that they take up and, in the sense widely misused in the literature, are excluders.
Response: Thank you for your critical comments. We removed the relevant content that caused misunderstanding. We introduce that recretohalophytes and salt-dilution halophytes are typical euhalophytes. According to Yuan et al. (2016), recretohalophytes actively secrete excessive salt to the surface of stems and leaves through salt glands or salt bladder, and according to Song et al. (2017), salt-dilution halophyte segregate salt ions to vacuole. And the details were shown on lines 46-48 of the revised version. And the number of the reference was [5] and [6].
[5] Yuan, F.; Leng, B.; Wang, B., Progress in Studying Salt Secretion from the Salt Glands in Recretohalophytes: How Do Plants Secrete Salt? Frontiers in Plant Science 2016, 7, 977.
[6] Song, J.; Shi, W.; Liu, R.; Xu, Y.; Sui, N.; Zhou, J.; Feng, G., The role of the seed coat in adaptation of dimorphic seeds of the euhalophyte Suaeda salsa to salinity. Plant Species Biology 2017, 32, 107-114.
(4)Line 59. Since the paper is about the mechanism of secretion of ions, the three hypotheses should be described in more detail. Summarise the three hypotheses and comment on how they might be distinguished experimentally.
Response: Thank you for your critical comments. We specifically describe three hypotheses and research supports. The osmotic potential hypothesis was firstly proposed in Limonium latifolium, supported by the ultrastructure of Distichlis spicata. It suggests that ions actively accumulate in salt glands, when the salt gland pressure reaches its highest, ions are discharged from the secretory pores of the salt gland cells through the microdroplets formed periodically to relieve pressure [16, 17]; As for similar to animal fluid transport hypothesis, is a transfer system, was proposed by Levering and Thomson (1971) [14]. Some studies support this hypothesis, salt secretion was significantly decreased after Na+-ATPase specific inhibitor (ouabain) treatment in Chloris gayana [18, 19]; As regards vesicle-mediated exocytosis hypothesis, proposed by Ziegler and Lüttge in 1967 [13], received support from L. bicolor. The study showed that electron-dense substances accumulated in the vesicles of salt glands [20, 21]. The results of Flowers et al. (2019) also supported this hypothesis, they collected flux data of Na+ and Cl- in halophytes, based on the membrane area and weight of the transport organs [22]. And the details were shown on lines 68-81 of the revised version.
(5) Lines 61-62. The authors do not mention recent reviews of Caperta et al (2020) where the origin of and function of glands within the Plumbaginaceae is discussed or, particularly relevant, the review of Flowers et al (2019), in which vesicular transport.
Caperta AD, Rois AS, Teixeira G, Garcia-Caparros P, Flowers TJ (2020) Secretory structures in plants: lessons from the Plumbaginaceae on their origin, evolution and roles in stress tolerance. Plant Cell and Environment 43 (12):2912-2931. doi:10.1111/pce.13825
Flowers TJ, Glenn EP, Volkov V (2019) Could vesicular transport of Na+ and Cl- be a feature of salt tolerance in halophytes? Annals of Botany 123 (1):1-18. doi:10.1093/aob/mcy164
Response: Thank you for your critical comments. We read and cited these studies. We briefly introduced the salt gland structure of the Limonium and L. bicolor by referring to the literature of Caperta et al. (2020) and Feng et al. (2014). We cited the reference of Flowers et al. (2019) to support that vesicle transport may be involved in salt gland secretion. And the details were shown on lines 50-54, 79-81 of the revised version. And the number of the reference are [1, 10, 22].
[1] Feng, Zhongtao; Sun, Qiuju; Deng, Yunquan; Sun, Shufeng; Zhang, Jianguo; Wang, Baoshan (2014). Study on pathway and characteristics of ion secretion of salt glands ofLimonium bicolor. Acta Physiologiae Plantarum, 36(10), 2729–2741. doi:10.1007/s11738-014-1644-3
[10] Caperta AD, Rois AS, Teixeira G, Garcia-Caparros P, Flowers TJ (2020) Secretory structures in plants: lessons from the Plumbaginaceae on their origin, evolution and roles in stress tolerance. Plant Cell and Environment 43 (12):2912-2931. doi:10.1111/pce.13825
[22] Flowers TJ, Glenn EP, Volkov V (2019) Could vesicular transport of Na+ and Cl- be a feature of salt tolerance in halophytes? Annals of Botany 123 (1):1-18. doi:10.1093/aob/mcy164
(6) Line 84 et seq. As this paper is about salt-secreting glands, the authors should include, at a minimum, concentrations of Na, K and Cl in the external solution on which the leaf disks were floated and in the secretions collected from the glands in order to show the glands were functioning in the way that might be expected/has been previously demonstrated.
Response: Thank you for your critical comments. We supplemented the results of the concentration of Na+, K+ and Cl- in secretion sap secreted from salt glands under control and salt treatment condition as Fig. S3. When the leaf discs were treated with 200 mM NaCl, the concentration of Na+, K+ and Cl- in secretion sap secreted from salt glands were 4.08 μM and 4.64 μM, but the concentration of Na+ and Cl- were 0.54 μM and 0.71 μM under the control, which were significantly lower than NaCl condition. While the concentration of K+ was 0.12 μM significantly decreased under NaCl condition compared with 0.55 μM under control condition (Fig. S3). And the details were shown on lines 193-199, 377-382, 636-639 of the revised version.
(7) Line 111 et seq. Provide an image of the droplets on the leaf surface.
Response: Thank you for your critical comments. Leaves and salt vesicles in Fig S1 are real images.
(8) Line 114. What was the size of discs? We've added the leaf disk size
Response: Thank you for your critical comments. The diameter of leaf discs is 1 cm. We added the size of leaf discs, and the details were shown on line 139 of the revised version.
(9) Line 173 Figure 2. This figure is not of very good quality and does no more than show proteins were present and could be separated. I suggest the figure is transferred to the supplementary data. The text (lines 166-175) should be adjusted as appropriate.
Response: Thank you for your critical comments. We have modified Fig. 2 to the attached Fig. S4 and the details were shown on lines 203 and 641-643 of the revised version.
(10) Line 202-203. “malate dehydrogenase (mitochondrion marker).” There are malate dehydrogenases other than in the mitochondria: e.g., Oh et al (2016).
Oh SJ, Gu DR, Jin SH, Park KH, Lee SH (2016) Cytosolic malate dehydrogenase regulates RANKL-mediated osteoclastogenesis via AMPK/c-Fos/NFATc1 signaling. Biochem Biophys Res Commun 475 (1):125-132. doi:10.1016/j.bbrc.2016.05.055
Response: Thank you for your critical comments. We agree that malate dehydrogenase is present in the mitochondrial matrix and cytoplasm, and we standardize our statement. We removed the reference to malate dehydrogenase as a mitochondrial marker enzyme.
(11)Line 242, 243. “liquid secreted from these salt glands contained a variety of proteins and polypeptides”: Why are there so many different polypeptides? No explanation is provided; nor is possible contamination from cut cells discussed.
Does the size of the disc (and so ratio of cut edge to surface area) affect the number of proteins found in the collected drops? Can droplets be collected from intact leaves to eliminate possible leakage from damaged cells?
Response: Thank you for your critical comments.
Firstly, we believe that salt gland secretion supports the vesicle transport hypothesis. In the process of vesicle formation, a large number of cytosolic proteins or functional proteins in salt secretion are included. Short-term salt stress may promote vesicle secretion, and vesicle secretion selectivity is low, so a large number of proteins and peptides will be loaded and secreted. And we believe that many immature protein fragments will be loaded into the vesicles and secreted out of the cell when vesicle transport is enhanced due to the low selectivity of vesicle transport. We suggest that the presence of peptides may be due to a sudden intensification of vesicular transport carrying a large number of immature protein fragments. And the details were shown on lines 262-270 of the revised version.
Secondary, according to the result of Yuan et al. (2013), the density of salt glands in the leaves of L. bicolor in our experiment was stable, and the area of our leaf discs in different treatment was the same (The diameter of leaf discs is 1 cm). We believe that the secretion efficiency of salt glands mainly affects the volume of secretory sap and the number of protein and the secretion efficiency of ions, while the area of leaf disc is not the key factor. We believe that the leaf-dices experiment is a relatively reasonable experiment at present, because we choose the leaves of the same leaf position to obtain the leaf-discs, which reduces the systematic error of the experiment. On the other hand, we use mineral oil to prevent the volatilization of the fluid, which is better to realize in vitro tissues.
Finally, we believe that, If the cells, including the salt glands and mesophyll cells, have an outflow of cell fluid due to cell injury, they will be released into the salt solution rather than secreting the side that covers the mineral oil. In order to visually demonstrate the relative activity of cells, we carried out Evans blue staining and found that the preparation process of leaf discs and salt stress treatment would not cause extensive leaf damage. We added Fig. S2 to show that in vitro plant damage caused by leaf disc experiment was not the main factor affecting salt gland secretion, and salt treatment did not aggravate plant cell damage. And the details were shown on lines 132-135, 189-193 of the revised version.
Yuan, F.; Chen, M.; Leng, B.Y.; Wang, B.S. (2013). An efficient autofluorescence method for screening Limonium bicolor mutants for abnormal salt gland density and salt secretion. South African Journal of Botany, 88, 110–117. doi:10.1016/j.sajb.2013.06.007
(12) Line es 243, 244. “whose number and variety sharply declined under salt stress compared to control conditions.”. Why should salt treatment decrease the number of proteins? This needs comment in the Discussion.
Response: Thank you for your critical comments. We have added a discussion of the reduction in the number and variety of proteins under salt treatment. We think that after 2 weeks of salt watering treatment, the halophyte L. bicolor had adapted to the salt environment, and the salt glands carried out normal salt secretion activities, while the remaining proteins and peptides under salt treatment might participate in the normal salt secretion function of the salt glands. And the details were shown on lines 262-270 of the revised version.
Minor comments
(13) Line 21 Clarify “number proportion”.
Response: We have revised the description. And the details were shown on lines 25 and 336 of the revised version.
(14) Line 38/9. “valorization”. Delete
Response: We have revised the description.
(15) Line 59 “have been” ‘were’.
Response: We have revised the description. And the details were shown on lines 17 and 66 of the revised version.
(16) Line 124 “rinsedtwice” ‘rinsed twice’
Response: We have revised the description. And the details were shown on lines 150 of the revised version.
(17) Line 157 “Salt crystals were excreted”. ‘Crystals’ were surely not excreted. but form after excretion of liquid containing salts.
Response: We have revised the description. And the details were shown on lines 185-187 of the revised version.
(18) Line 206: Figure 4. Spelling of “peroxisome” perooxisome’.
Response: We have revised the figure. And the detail was shown in revised Fig. 3.
(19) Line 208 “saps”; ‘sap’.
Response: We have revised the description. And the details were shown on lines 229, 235, 611, 615 of the revised version.
Reviewer 2 Report
The paper presents interesting results that contribute to the advancement of the knowledge of the secretion mechanism by salt glands of L. bicolor.
I consider that the quality of figure 4 can be improved. In figure 2 the red colour in (c) is not visible. In the discussion part, the conclusions could be highlighted more clearly.
Author Response
Dear Editor and Reviewers,
Thank you very much for your suggestions and critical comments on our paper submitted to International Journal of Molecular Sciences. We have revised the manuscript according to the comments and suggestions. Attached, please find the revised version. The main revised parts are marked in blue in the paper. The following is a detailed explanation how we have complied with the reviewers’ suggestions.
Comments and Suggestions for Authors:
The paper presents interesting results that contribute to the advancement of the knowledge of the secretion mechanism by salt glands of L. bicolor.
I consider that the quality of figure 4 can be improved. In figure 2 the red colour in (c) is not visible. In the discussion part, the conclusions could be highlighted more clearly.
Response: Thank you for your critical comments. We have re-edited the figures in the article and increased the figures resolution, the details were shown in the revised Fig. 3. And we modified Fig 4 to Fig 3, modified Fig. 2 to Fig. S4. And we didn't find the red color in Fig. 2, but we checked all the pictures and markers to make sure they were clear.
In the discussion, we add to the view that the reason of the number of proteins and peptides decreases under salt stress. We believe that the presence of peptides of secretion sap under control condition may be due to a sudden enhancement of vesicular transport carrying a large number of immature protein fragments. But under 2 weeks salt condition, the L. bicolor could adapt the activity of salt glands to secret Na+ and Cl-. We conclude that the decrease in the number of proteins and peptides is related to the adaptation of the salt gland secretion salt of the halophytes L. bicolor under salt treatment, and the remaining proteins and peptides after 2 weeks of salt treatment may be involved in the normal salt secretion under salt stress. And the details were shown on lines 262-270 of the revised version.
In order to better explain our experimental results, we add the protein species in the secretory fluid in the first section of the discussion. And the details were shown on lines 271-274 of the revised version.
According to the suggestion, we talked about the relationship between ions and proteins in the secretion sap. Under salt treatment after two weeks, the Na+ and Cl- concentration in secretion sap significantly increased, compared with the amount of protein and peptides less (Fig.2 and Fig.S3), we speculated that, the main function of salt gland is secreting ions, and this kind of ion and protein secretion balance is how to control under salt condition, whether there are proteins regulate the salt gland secretion, we do not yet know. And the details were shown on lines 377-382 of the revised version.
Reviewer 3 Report
This manuscript describes a proteomic study of salt gland-secreted sap from L. bicolor. The secretion liquid of leaf discs from plants watered with high salt solution and normal nutrient solution was collected to extract proteins for SDS-PAGE and LC-MS analysis. Unexpectedly, the authors found that high salt treatment resulted in a decrease in the number and the amount of secreted proteins. The authors also found that many identified proteins involved in vesicular transport, energy production, and material metabolism, indicating that vesicle transport plays a role in salt gland secretion. The scientific goal and potential implication of this study is significant, but the experimental design and data interpretation need improvement. I recommend publication upon addressing the concerns listed below.
Major concerns:
1. The rationales of the experimental setup should be more clearly stated in the results section. What exactly are the factors controlled by the control group? Do the authors intend to compare the secreted proteomes from plants before and after adaptation to high salt conditions?
2. What, if any, protein concentration measurement and normalization were performed during mass spectrometry-based proteomic experiments? Do the samples from high salt treatment group contain sufficient protein mass?
3. Many of the analysis in the manuscript relies on a comparison of the number proportion of proteins related to certain pathways between normal and treated samples (e.g., line 305 - 307), which is not meaningful as the total number of identified proteins in high salt group is only 16.
Minor concerns:
1. The protein number in Figure 3(a) is not meaningful. This figure should be re-drawn as a boxplot.
2. The color of text annotations (‘other’) in Figure 4 does not match the color of the corresponding pie fraction. The initial character of annotations should be consistently capitalized.
3. Peroxisome was misspelled in Figure 4.
Author Response
Dear Editor and Reviewers,
Thank you very much for your suggestions and critical comments on our paper submitted to International Journal of Molecular Sciences. We have revised the manuscript according to the comments and suggestions. Attached, please find the revised version. The main revised parts are marked in blue in the paper. The following is a detailed explanation how we have complied with the reviewers’ suggestions.
Comments and Suggestions for Authors:
Comments and Suggestions for Authors
This manuscript describes a proteomic study of salt gland-secreted sap from L. bicolor. The secretion liquid of leaf discs from plants watered with high salt solution and normal nutrient solution was collected to extract proteins for SDS-PAGE and LC-MS analysis. Unexpectedly, the authors found that high salt treatment resulted in a decrease in the number and the amount of secreted proteins. The authors also found that many identified proteins involved in vesicular transport, energy production, and material metabolism, indicating that vesicle transport plays a role in salt gland secretion. The scientific goal and potential implication of this study is significant, but the experimental design and data interpretation need improvement. I recommend publication upon addressing the concerns listed below.
Major concerns:
- The rationales of the experimental setup should be more clearly stated in the results section. What exactly are the factors controlled by the control group? Do the authors intend to compare the secreted proteomes from plants before and after adaptation to high salt conditions?
Response: Thank you for your critical comments. We added some experimental setup principles. And the details were shown on lines 189-193, 227-228 of the revised version.
Our variable is salt treatment. Our experiment focused on whether the salt glands secreted ions and also secreted proteins, and whether the protein secretion would be affected by the salt environment. Therefore, we extracted proteins in secretion sap treated with 0 mM and 200 mM NaCl 2 weeks and conducted component analysis, focusing on the influence of the salt environment.
Yes, we compared the proteome of 200 mM NaCl treatment with that of the control group after 2 weeks. We found that under the control condition, there were more types and quantities of proteins and peptides, while under NaCl condition, the proportion of proteins changed, for example, the proportion of proteins related to photosynthesis and respiration in the total protein increased significantly under salt stress.
- What, if any, protein concentration measurement and normalization were performed during mass spectrometry-based proteomic experiments? Do the samples from high salt treatment group contain sufficient protein mass?
Response: Thank you for your critical comments, each of our samples with three biological replicates, was uniformly diluted with protein concentrations measured by Bradford method before mass spectrometry with three biological replicates.
- Many of the analysis in the manuscript relies on a comparison of the number proportion of proteins related to certain pathways between normal and treated samples (e.g., line 305 - 307), which is not meaningful as the total number of identified proteins in high salt group is only 16.
Response: Thank you for your critical comments. The focus of our experiment is to find out whether there are proteins and peptides in the secretion sap of salt gland, and whether the secretion activity is affected by the salt environment. The proportion we focus on is the change of protein composition under different treatments, especially after salt treatment. According to our results, the number of salt-treated proteins secreted was significantly reduced, and only 16. These remaining proteins may participate in the salt secretion process of salt glands or respond to salt condition. And among of these proteins, enzymes related to photosynthesis and respiration accounted for 50% of the total proteins, so we speculated that related enzymes might provide a large amount of energy in the secretion of salt glands under salt environment.
Minor concerns:
- The protein number in Figure 3(a) is not meaningful. This figure should be re-drawn as a boxplot.
Response: Thank you for your critical comments. We thought carefully about your suggestion and tried to redraw the boxplot. However, the error of the protein data of salt treatment was large, which could not reflect the experimental results aesthetically, so we still chose the original picture. In addition, we also explained the reason for the error. The plants respond quickly to salt stress through functional peptides when suddenly exposed to salt stress. Since vesicle transport is determined by the local ion concentration near the vesicle and the charge of the vesicle membrane, the complement of secreted proteins appeared to be poorly reproducible. In the revised version, we changed Fig. 3 to Fig. 2.
- The color of text annotations (‘other’) in Figure 4 does not match the color of the corresponding pie fraction. The initial character of annotations should be consistently capitalized.
Response: Thank you for your critical comments. We have redrawn the figure and improved the quality of the figure and matched the appropriate color. In the revised version, we changed Fig. 4 to Fig. 3.
- Peroxisome was misspelled in Figure 4.
Response: Thank you for your critical comments. In the revised version, we changed the Fig. 4 to Fig. 3. And we changed the spelling of the word “peroxisome” and improved the quality of Fig. 3.
Round 2
Reviewer 3 Report
The revised manuscript addressed my concerns.